# Epstein–Barr Virus—Oral Bacterial Link in the Development of Oral Squamous Cell Carcinoma

**DOI:** 10.3390/pathogens9121059

**Published:** 2020-12-18

**Authors:** Daniela Núñez-Acurio, Denisse Bravo, Francisco Aguayo

**Affiliations:** 1Laboratory of Oral Microbiology, Faculty of Dentistry, University of Chile, Santiago 8380492, Chile; danynuac@gmail.com; 2Laboratory of Oncovirology, Virology Program, Faculty of Medicine, Institute of Biomedical Sciences (ICBM), University of Chile, Santiago 8380000, Chile; 3Advanced Center for Chronic Diseases (ACCDiS), Faculty of Medicine, University of Chile, Santiago 8380000, Chile

**Keywords:** oral squamous cell carcinoma, Epstein–Barr, *Porphyromonas gingivalis*, epithelial carcinogenesis

## Abstract

Oral squamous cell carcinoma (OSCC) is the most common type of oral cancer. Its development has been associated with diverse factors such as tobacco smoking and alcohol consumption. In addition, it has been suggested that microorganisms are risk factors for oral carcinogenesis. Epstein–Barr virus (EBV), which establishes lifelong persistent infections and is intermittently shed in the saliva, has been associated with several lymphomas and carcinomas that arise in the oral cavity. In particular, it has been detected in a subset of OSCCs. Moreover, its presence in patients with periodontitis has also been described. *Porphyromonas gingivalis* (*P. gingivalis*) is an oral bacterium in the development of periodontal diseases. As a keystone pathogen of periodontitis, *P. gingivalis* is known not only to damage local periodontal tissues but also to evade the host immune system and eventually affect systemic health. Persistent exposure to *P. gingivalis* promotes tumorigenic properties of oral epithelial cells, suggesting that chronic *P. gingivalis* infection is a potential risk factor for OSCC. Given that the oral cavity serves as the main site where EBV and *P. gingivalis* are harbored, and because of their oncogenic potential, we review here the current information about the participation of these microorganisms in oral carcinogenesis, describe the mechanisms by which EBV and *P. gingivalis* independently or synergistically can collaborate, and propose a model of interaction between both microorganisms.

## 1. Introduction

Oral cancer develops in the lips, cheeks, floor of the mouth, mobile tongue, and hard palate, buccal alveolar bone, retromolar trigone, and soft palate [1]. In particular, oral squamous cell carcinoma (OSCC) is the most frequent histological type (90%), followed by salivary gland malignancies, sarcomas, and melanomas (6%), and Hodgkin lymphomas (4%) [2]. Lip and oral cavity cancer is the fourth most common cancer and the sixth most common cause of cancer deaths in low- and middle-income countries [3]. In 2018, the International Agency for Research on Cancer (IARC), reported an estimated 354,864 new cases of the lip and oral cavity cancer worldwide, representing 2.1% of all cancers [4]. Highly prevalent countries are India, Pakistan, Afghanistan, Bangladesh, Sri Lanka, Bhutan, Nepal, Iran, Maldives, and Papua New Guinea [4,5,6]. In Latin America, the highest incidence and mortality rates are reported in Brazil followed by Mexico, Argentina, Colombia, and Cuba [7,8]. Oral cancer is identified in people with a mean age of 50 years, and is three times more frequent in men than in women [9,10,11]. Five-year survival rates of 54% are reported, primarily due to late-stage diagnosis [11,12]. The clinical presentation of this cancer is an ulcerated lesion in the oral cavity, and the presence of mobile teeth, bleeding, pain, and numbness in the mouth [1]. Leukoplakia, erythroplakia, oral lichen planus, and oral submucous fibrosis are identified as precancerous lesions [13].

Tobacco smoking (TS) and alcohol consumption are often considered as the major risk factors for oral cancer [14,15]. Among smokers who never drink alcohol, there is a two-fold risk estimate for oral cavity and oropharyngeal cancer, which increases with frequency and duration of smoking, while drinking among non-smokers seems unrelated to oral cancer risk [16,17]. The risks of oral cancer increase at least five-fold in those subjects who both smoke tobacco and drink alcohol [15,16,17]. In addition, a variety of suspected risk factors such as teeth or dental/denture trauma, poor oral hygiene, sexual behavior, malnutrition as well as low fruit and vegetable diets, and genetic factors, have been proposed for the development of oral cancer [18,19,20]. However, some subjects are unexposed to these factors and present malignant oral lesions [21,22]. Thus, additional factors are potentially involved in oral cancer, with a suggested role of some viruses and oral microbiome, especially the bacteriome [23,24]. For instance, high-risk human papillomaviruses (HR-HPV), particularly HPV16 and 18, are detected in 24–56% of oral cancers worldwide [25,26,27]. Besides, Epstein–Barr virus (EBV) has been studied for its association with the development of OSCCs [28,29] and inflammatory oral diseases. Periodontitis is described as the most common of them and it could contribute to the triggering of cancer [30]. Consequently, the purpose of this review is to describe the role of EBV and the periodontitis-associated bacteria, *P. gingivalis*, in the development of oral cancer, as well as to revise the current literature regarding the interaction between these two microorganisms.

## 2. Epstein–Barr Virus in Oral Cancer

### 2.1. Epstein–Barr Virus as a Risk Factor 

EBV infection is present in 90% of human population [31,32,33]. The virus is transmitted through saliva, being usually asymptomatic during childhood; however, if the infection occurs in adolescence, it can lead to infectious mononucleosis (IM) [33]. In 1997, EBV was declared as the causal agent of nasopharyngeal carcinoma by the IARC [34]. EBV is also related to the development of various types of cancers such as Hodgkin lymphoma (HL), natural killer cell lymphoma/T lymphocytes (NK/T), gastric cancer (GC), and lymphoproliferative disorders (LPDs) [35,36,37,38,39]. 

The gold standard for EBV detection is in situ hybridization (ISH) for EBERs. Additionally, polymerase chain reaction (PCR) for EBNA or LMP1 sequence detection is frequently used [40]. EBV genomes have been detected in approximately 45% of OSCCs by PCR and ISH [29,41,42] and in 80% by transcriptomic approaches [29,43]. In fact, She et al. showed an association between EBV infection and increased risk of OSCC presenting an OR of 5.03 [28]. Moreover, a meta-analysis that considered 53 studies from 1990 to 2019 shows evidence that EBV-infected individuals have a 2.5 increased risk for developing OSCC [29]. 

### 2.2. Epstein–Barr Virus Replication Cycle in Epithelial Cells 

EBV is a linear double-stranded DNA virus that belongs to the Herpesviridae family, gammaherpesviruses subfamily, and Lymphocryptovirus genus, with tropism for B-cells and epithelial cells. The primary EBV infection is probably initiated in oral epithelial cells in a lytic form, and the virus subsequently infects B-cells, where it usually persists in a latent form [44]. In epithelial cells, the virus entry occurs by fusion of the cell membrane through the viral fusion complex gH/gL with the αvβ5, αvβ6, and αvβ8 integrins [45]. Additionally, the interaction with the cellular receptor EphA2 is suggested to be necessary for virus entry [46]. Then, the capsid is disassembled, the tegument proteins are released into the cytoplasm, and the viral genome is transported to the nucleus [44]. EBV BNRF1 protein interacts with the host nuclear protein Daxx, disrupting Daxx/Atrx chromatin remodeling complex to facilitate the initiation of viral gene transcription [47], which is a temporally regulated process that encodes over 85 gene products. The immediate-early (IE) proteins, BZLF1, and BRLF1 act as transcription factors promoting the expression of early genes, including those which encode viral DNA synthesis [48]. Such early gene products include the viral DNA polymerase catalytic subunit (BALF5), polymerase accessory factor (BMRF1), ssDNA-binding protein (BALF2), and components of the helicase/primase complex (BBLF4, BSLF1-BBLF2/3) [49]. The replication cycle ends with late genes expression, including virion structural components such as BLRF2 (tegument protein) and the DNA packaging proteins BXLF2 (gH), BALF4 (gB, gp110, gp125), and BKRF2 (gL, gp25), among others [49]. The lytic cycle concludes with virions production, which can infect more susceptible cells. Thus, the virus can colonize salivary glands, lymphoid tissues, and epithelial cells of the oropharynx circulating mainly between peripheral blood and the Waldeyer’s ring, where naïve B-cells are infected [50]. In B-cells, the virus can establish a latent cycle with a restricted and highly regulated gene expression program [51]. 

### 2.3. Epstein–Barr Virus and the Activation of Mechanisms Associated with Oral Cancer

EBV has been linked to the development of epithelial cancers such as nasopharyngeal carcinoma and gastric cancer [39,52]. Even though EBV infection in epithelial cells is generally lytic, it has been observed that a latency state can be established in cells that show prior genomic DNA damage, like cyclin D1 overexpression [53,54,55]. A study showed that latent forms of EBV promote cell proliferation in some epithelial and oral cells, inducing significant E-cadherin and ZO-1 down-regulation [40]. In fact, viral latency allows for sustained expression of viral oncogenes, while it avoids immune detection and cytopathic effects from the viral replicative phase [53]. EBV-positive squamous epithelial cells show latency I, characterized by expression of EBERs1/2, EBNA1, and BARTs and, latency II defined by EBERs1/2, EBNA1, LMP1, and LMP2 expressions [55]. In addition, the expression of some lytic genes has been identified. BamH1-A Reading Frame-1 (BARF-1), which encodes a 220 amino acid protein, is considered an EBV epithelial oncogene [56]. It is detected in 90% of EBV-positive carcinomas during latency and is absent in EBV-positive lymphomas [57,58,59]. Importantly, BARF-1 is secreted as a hexameric complex of 31–33 kDa [56], whose expression is triggered by the IE protein, BZLF1, during lytic cycle activation and promoted by the cell differentiation factor ΔNp63α in epithelial cells [60]. The first 54 amino acids at the N-terminus, including the membrane-spanning domain, were found to be responsible for the cellular antiapoptotic protein Bcl-2 upregulation, mediating escape from senescence upon serum deprivation and mediating growth in soft agar [61]. Additionally, it has immunomodulatory functions by inhibiting macrophage colony-stimulating factor (M-CSF) [62]. BARF-1 exhibits homology with the extracellular domain of the M-CSF receptor (M-CSFR). Hence, it allows its binding with M-CSF and inhibits its activity, which is related to the regulation of viability, proliferation, and differentiation of monocytes and macrophages [62]. BARF-1 exhibits immortalization and transformation capabilities in cooperation with the H-Ras gene [63], is involved in the inhibition of apoptosis through increased Bcl-2 expression, and is an activator of the NF-κB/CyclinD1 pathway [64,65]. Moreover, BARF1 stimulates cell growth and survival when expressed in epithelial cells [66]. Unfortunately, according to our knowledge, BARF1 levels have not been evaluated in EBV positive oral OSCCs. Thus, this point warrants further investigations. 

Latent membrane protein-1 (LMP1), another EBV-encoded oncogenic protein, has also been associated with oral carcinomas. The prevalence of LMP1 protein in EBV-associated OSSCs ranged from 10.7 [67] to 59.6% [68]. LMP1 triggers multiple signaling pathways, including PI3K (Phosphoinositide 3-kinase) /Akt (protein kinase B) and MAPK (Mitogen-Activated Protein Kinases) [69]. Moreover, this protein contributes to epithelial cell transformation through activation of transcription factors such as NF-κB and Stats [70], and the activation of proteins involved in adhesion and invasion such as E-cadherin and MMP9 [71]. It has been demonstrated that LMP1 induces increased cell growth in soft agar with enhanced migration in EBV positive NPC cells, fibroblasts, non-tumor keratinocytes, and kidney epithelial cells [71,72,73], requiring the activation of both PI3K/Akt and canonical NF-κB signaling [74]. 

## 3. Oral Bacterial in the Development of Cancer 

The oral cavity is one of the anatomical sites that contain more diverse microbiota, hosting at least 687 species [75]. Both pathogenic and mutualistic bacteria coevolve together to maintain oral homeostasis [76]. However, under certain conditions, such as smoking, obesity, stress, diabetes, or even the presence of specific microorganism, the balance of the oral ecosystem is altered, allowing an increase in the number of pathogenic bacteria that produce damage in the tissue because of the expression of their virulence factors and the consequent host immune response [77]. This is called the theory of polymicrobial synergy and dysbiosis (PDS), which proposes that the loss of homeostasis is caused by keystone pathogens (mostly pathogenic bacteria) that communicate with the accessory pathogens, acting synergistically to support the virulence of the disease-associated organism and facilitating the progression toward pathogenesis [78]. In this context, periodontitis fits the PDS concept because it is a dysbiotic disease characterized by chronic inflammation caused by an imbalance between subgingival microbiota, mostly mediated by the Gram-negative key pathogen *Porphyromonas gingivalis*, and the host immune response, leading to the destruction of supporting tissues of the teeth [77]. *P. gingivalis* show high relative abundance in periodontitis patients; however, this bacterium can also be present in a smaller proportion in healthy patients [79]. Therefore, its association with periodontitis is proposed to be due to the change in its relative abundance within the subgingival microbial community [80,81]. Additionally, its participation in periodontitis is determined by interaction with other bacteria. A synergistic effect on virulence between *P. gingivalis* and *Fusobacterium nucleatum* is reported, causing an increase in the internalization capacity of *P. gingivalis* into epithelial cells [82,83]. Furthermore, interaction has been described between *P. gingivalis* and *Aggregatibacter actinomycetemcomitans*, which protect *P. gingivalis* from the H_2_O_2_ produced by *Streptococcus sanguinis*, favoring their survival and growth [84].

Due to the damage, a nutrient-rich exudate is produced and further favors the growth of pathogenic bacteria. As this process is perpetuated, severe chronic inflammation is produced. Periodontitis has not only been related to local inflammation but also systemic inflammation due to dysregulation in host immune response. It represents a strong increased risk factor for the development of some diseases, such as coronary heart disease (CHD) [85]. It has been reported that several inflammatory mediators are released into the bloodstream during periodontitis such as PCR, metalloproteases, and prostaglandins [86]. Oxidative stress, through nitric oxide (NO), has been determined to be one of the main features in the pathogenesis of both periodontitis and CHD. NO is one of the important mediators that regulate the function and vasodilatation of the endothelium. The inflammatory mediators have been described to decrease the production of NO, which negatively impacts vascular endothelial cells, whose deterioration determines endothelial dysfunction, vasodilatation, and CHD [86]. Some markers of endothelial function have been correlated with periodontitis, such as soluble urokinase-type plasminogen activator receptor (suPAR) and asymmetric dimethylarginine (ADMA), which functions as an inhibitor of the NO metabolism [85,87]. The detection of these mediators could be used as a biomarker of oral health status and CHD. Moreover, the detection of high levels of antibodies against *P. gingivalis* and *Aggregatibacter actinomycetemcomitans*, a low-abundance Gram-negative pathogen highly related to a silent but aggressive periodontitis, have been associated with early stages of cardiovascular diseases and correlated with the degree of severity of periodontitis [88]. Hence, antibody measurement against these periodontitis-associated pathogens may be useful to support diagnosis and prevent the progression to more severe stages of the disease. 

On the other hand, local inflammation perpetuated in the oral cavity, associated with periodontitis progression, potentially leads to the development of oral cavity tumors. An association between periodontal disease and orodigestive cancers (oral, esophageal, gastric, colonic, and pancreatic) has been well established [89].

Currently, evidence suggests that individuals diagnosed with periodontitis are at high risk of developing oral cancer (HR 1.79, 95% CI: 1.42–2.25) [90]. The participation of periodontitis in the progression of OSCC can be understood because the inflammatory mediators produced during periodontitis can promote the destruction of soft and hard tissues of the mouth, activating the oral osteoclast. This favors tumor cell invasion to adjacent tissues, resulting in malignant transformation of lesions leading to OSCC. Apart from this, the participation of periodontitis in COCE can also be explained because periodontal bacteria can directly affect the transformation of cells [91]. The participation of several oral bacteria present in high loads in the periodontal pocket is mentioned. It has been shown that oral bacteria activate inflammatory pathways associated with various stages of cell transformation [89]. Intriguingly, persistent exposure of *P. gingivalis* to oral epithelial cells promotes changes in host cells associated with hallmarks of carcinogenesis. It has been shown that *P. gingivalis* causes changes in cell morphology, increases the proliferation of cells, and increases their migratory and invasive properties [92]. *P. gingivalis* stimulates the growth of primary gingival epithelial cells (GECS), which, in the early stage of periodontal diseases, regulate the production of reactive oxygen species (ROS) [93]. ROS mediate activation of pro-oncogenic signaling pathways that later facilitate cancer progression, angiogenesis, and survival [94]. Additionally, ROS can generate modifications in the nitrogenous bases of DNA, inducing genome instability, and mutations [95]. Furthermore, *P. gingivalis* increases the proliferation of primary fibroblasts of the periodontal ligament and promotes the proliferation of immortalized gingival cells [96]. Finally, *P. gingivalis* induces GEC migration in a Zeb1-dependent manner, which is an activator of epithelial-mesenchymal transition (EMT) [97].

*P. gingivalis* virulence factors have a direct role in the promotion of these properties. The lipopolysaccharide (LPS) activates immune response through toll-like receptors (TLR-2, TLR-4) associated with downstream activation of inflammatory pathways such as PI3K/Akt, JAK/STAT, and production of IL-1β in the host cells [98,99,100]. In particular, the O-antigen region of the LPS contributes to the inhibition of apoptosis and induces the proliferation of GEC [101]. On the other hand, the capsule is associated with the evasion of the immune system and the promotion of the bacteria’s survival in host cells [102,103]. Moreover, the fimbriae (FimA and Mfa1) not only allow adherence to the cell surface and promote aggregation, but also induce the expression of cytokines like IL-1, IL-6, and TNF-α from monocytic and epithelial cells mediated by TLR-2 [104,105]. Finally, the gingipains (RgpA, RgpB, and Kgp) are strong proteases that degrade proteins of the complement system, cytokines, integrins, and collagen, producing severe damage of the cell-to-cell contacts and detachment of epithelial cells from connective tissues of the gingiva [106,107]. Together, the persistence of the bacteria in the host cell, the tissue degradation by gingipains, and the activation of immune effectors by the LPS and fimbriae suggest the participation of *P. gingivalis* in the severe inflammation processes. Intriguingly, the activation of inflammatory mediators is closely related to the activation of oncogenic pathways by intermediators, such as NF-κB; hence, periodontal diseases and oral cancer can be mediated by *P. gingivalis* in many ways.

## 4. Signaling Pathways in Oral Cancer

Oral carcinogenesis is a multistep and multifocal process that involves a complex interaction network [108]. Many studies have identified overexpressed or mutated genes related to oral tumorigenesis, among which are those involved in proliferation (PI3K/Akt/mTOR, NOTCH, H-ras), apoptosis regulation (Bcl2, Bax), cell-cycle control (p53, cyclin D, CDKN2), leading to increased migration and invasion (Zeb, Vimentin, Slug, Snail) (Figure 1) [109,110,111].

P53, a tumor suppressor protein, acts as a regulator of DNA synthesis and is implicated in the expression of proteins involved in cell cycle arrest, DNA repair, and apoptosis. Among these P53-regulated proteins are retinoblastoma protein (pRb), p21 (cyclin-dependent kinase inhibitor 1), and Bax (Bcl-2 associated X protein) (Figure 1) [112,113]. More than 50% of all primary head and neck squamous cell carcinomas (HNSCCs) show TP53 alterations [110,111,114]. Missense mutations in the DNA-binding domain are the most common alterations in the TP53 gene, associated with a significantly decreased survival [115]. The C: G to A: T transitions at codons R248, R273, G245, R175, R282, and H179 are the most prevalent hotspot mutations in HNSCCs [114,116]. Aside from the loss of function, a gain of function is described in p53 mutants leading to cell invasion, migration, proliferation, and drug resistance [117] because of changes in DNA binding properties and altered protein–protein interactions [118]. 

Pathway analysis of the genomics data indicated that 70% of OSCCs had altered genes in the RTK/MAPK/PI3K pathway [110] (Figure 1). Epidermal growth factor receptor (EGFR) is overexpressed in 40–60% of the OSSCs [119,120,121], resulting in a poor prognosis. Its activation is associated with the malignant phenotype, apoptosis inhibition, and increased metastatic potential [122,123]. Genetic variations in EGFR may alter protein function, contribute to tumor formation, and possibly alter the therapeutic efficacy of EGFR inhibitors. The most common mutation described is the change in the number of CA simple sequence repeats (normally 9–23 repeats) present in the first intron of the EGFR gene, [124]. If there are short CA repeats, a higher EGFR mRNA expression compared to long CA repeats occurs. Hence, cells with fewer CA repeats are more resistant to EGFR inhibitors such as erlotinib or cetuximab and prone to active downstream oncogenic pathways [125].

H-ras (Harvey-Ras) transduces the growth signal from EGFR to intracellular effectors through MAPK, c-Jun N-terminal kinase (JNK), and PI3K/Akt pathways, which regulate normal cell proliferation (reviewed in [126,127]. H-ras is significantly overexpressed in oral carcinomas (60%) compared to normal oral tissues (41.5%), which means inadequate downstream signal transduction [128]. In oral tumors, 35% of H-ras mutations have been reported [129]. The major mutations described are T > A and T > C in H-ras codon 61, causing the mutant protein to lose its ability to exchange GTP with GDP, thus remaining constitutively activated [129,130]. It has been described that overexpression of miR182, a non-coding RNA that acts as a tumor suppressor, sustains Ras–MEK–ERK signaling-pathway activation, promoting cell proliferation, cell-cycle progression, colony formation, and invasion capacity in OSCC cell lines [131]. 

PI3K/Akt/mTOR pathway is the most frequently mutated pathway (11–30%) in HNSCC patients [132,133], in contrast to JAK/STAT or the MAPK pathways (8–12%) [133] (Figure 1). PI3K is an intracellular signal transducer with the capability of activating Akt protein. Akt plays a key role in multiple cellular processes, as it regulates cell growth through its effects on mTORc signaling (Figure 1) [134]. Furthermore, Akt contributes to cell proliferation via phosphorylation of the CDK inhibitors p21 and p27. Akt also regulates NF-κB signaling by phosphorylating IKKα and Tpl2 [135]. Similarly, it has action over cell survival through direct inhibition of pro-apoptotic proteins like Bad or inhibition of pro-apoptotic signals generated by transcription factors such as FoxO [136]. PIK3CA is the most mutated gene in the HNSCC-PI3K mutational profile [133,137]. Different missense mutations are described in exon 20, exon 9 producing a change from A3140G to H1047R and A1634G to E545G, respectively [138]. PIK3CA mutant has been involved in cervical lymph nodes or IV stage patients, suggesting that PIK3CA mutations may be a late event in the progression of OSCC [110]. The tumor suppressor phosphatase and tensin homolog (PTEN), which inhibits Akt, is described as frequently mutated or lost in oral tumors causing continuous downstream activation of Akt [139,140].

NOTCH signaling acts as a transcriptional activator, and its function is to communicate signals upon binding to transmembrane ligands expressed on adjacent cells [141] (Figure 1). NOTCH1 is a frequently mutated gene in oral cancer (25%) among all four human Notch paralogs, submitting missense mutations (G310R, D352G, R365C, T1014M, C1383Y, and Q1957P) [110]. Upregulation of Notch ligands as JAG1 and JAG2 has been identified in OSCCs producing deregulation in the pathway activity and increased cell proliferation [142]. Moreover, it has been described that continuously activated notch signaling mediates tumor metastasis in OSCC cells through its association with Snail-mediated EMT progression [143].

Constitutive overexpression of cyclin D1 has been detected in oral tumors, suggesting the implication of this protein in tumorigenesis [144] (Figure 1). Cyclin D1 is an important regulator of cell cycle progression and functions as a transcriptional co-regulator. Its expression is regulated by EGFR, NF-κB, STAT3, and β-catenin; thus, its alteration implies a cyclin D1 deregulation. It has been shown that pSTAT3 is more associated with cyclin D1 overexpression in OSCC samples [144]. Nonsense and frameshift mutations are observed in invasive tumors in p16 (CDKN2), which is an inhibitor of cyclin D [145]. A > 5-fold downregulation of CDKN2A is reported, due to a copy number deletion with a nonsense mutation that produces a stop/loss of function of the gene, which results in a loss in cell cycle regulation and deregulated proliferation [146].

Modification in apoptosis is also described in oral tumorigenesis. Bcl-2 family proteins include antiapoptotic proteins such as Bcl-2 and Bcl-xl and pro-apoptotic proteins like Bax and Bad [147] (Figure 1). The early phase of epithelial carcinogenesis is attributed to Bcl-2 protein overexpression, resulting in an alteration of programmed cell death, with the persistence of cells that fail to die [148,149]. Moreover, an alteration in pro-apoptotic proteins promotes the inhibition of apoptosis. Mutation as a single nucleotide polymorphism (SNP) G-248A [150] and methylation in the promoter and coding regions of the Bax gene have been reported in HNSCC, associated with reduced expression from 76.9% in normal samples to 57.8% in primary oral tumors [151]. 

The proliferation and uncontrolled growth allow adaptive advantages over the surrounding cells and generate an expression of new factors like a set of chemokines, CXCL1, CXCL8 triggering signaling cascades, and activating immune responses [152]. This microenvironment promotes cell adhesion loss and facilitates EMT [153]. In the EMT, the expression of transcription factors such as Zeb, Vimentin, Slug, and Snail occurs by inhibition of E-cadherin expression and promotion of N-cadherin, causing the loss of maintenance of cell junctions and cellular polarity [154]. Moreover, MMP2, MMP9, MMP13 (matrix metalloproteinases) are expressed by tumor cells and are involved in collagen, and extracellular matrix protein degradation [155].

## 5. Signaling Pathways Involved in EBV-Bacteria Interactions in the Oral Cavity

The microbial communities in the oral cavity exert complex interactions that modify the behavior of the microbiome and the host cell responses. In this context, recent studies confirm that EBV and *P. gingivalis* are increased in patients with chronic periodontitis, suggesting that EBV infection could contribute to the etiopathogenesis of periodontitis [156,157,158]. In fact, the co-existence of *P. gingivalis* and EBV in Japanese chronic periodontitis patients were detected in 80% of sites with probing pocket depths (PPD) greater than 5 mm. The higher level of EBV and *P. gingivalis* coinfection was detected in chronic periodontitis and aggressive periodontitis patients when compared to healthy subjects [159]. In the same context, another study confirmed a higher proportion of *P. gingivalis* detected in EBV-positive subjects and it was demonstrated that stimulation with *P. gingivalis* supernatant results in EBV reactivation in an EBV-positive B-lymphoid cell line (Raji; Japanese Collection of Research Bioresources) [160]. 

In the signaling context, *P. gingivalis* and EBV independently have the ability to activate tumorigenic pathways. The activation of the EGFR/PI3K/Akt pathway is altered in EBV-positive tumors and occurs in *P. gingivalis* infection [110,161]. This pathway is key in oncogenic development because it inhibits intrinsic apoptosis proteins (Bad and Bax) and promotes NF-κB activation, which increases the transcription of anti-apoptotic genes [135,162]. Moreover, Akt leads to the activation of GSK-3β, FOXO, and cyclin D1 to enhance proliferation [163]. Finally, mTOR signaling impacts the survival, protein synthesis, and cytoskeletal organization of cells [164]. Hence, as PI3K/Akt pathway is activated by both EBV and *P. gingivalis* and widely described to be altered in OSCC, it could be suggested that it plays an important cooperative or collaborative role between both microorganisms in the development of oral cancer. However, this possible role remains to be investigated in molecular detail. In Figure 2, we propose a model of this possible EBV/*P gingivalis* interaction. Additionally, studies propose that *P. gingivalis* induces the EBV lytic switch [165]. BZL1 gene promoter (immediately early gene) is enveloped by histones and its activation requires the inhibition of a deacetylated histone (HDACs) [166]. Butyric acid, a short-chain fatty acid is a metabolic waste component of *P. gingivalis* that promotes the inhibition of HDACs [167]. Therefore, it generates the acetylation of histones, the activation of the BZL1 gene promoter, and finally the activation of the EBV abortive lytic cycle [168]. The concentration of butyric acid is reported to increase with the progression of periodontal disease, suggesting that it may promote a severe state of the disease [169]. In fact, the abortive lytic cycle, in which a limited set of viral products are expressed without viral maturation, is prominent in epithelial carcinogenesis [170]. Among the viral genes expressed in this abortive lytic cycle are the immediately early and early ones, but not the structural genes, and, hence, no new viral particle is produced [171]. In the EBV lytic cycle, the expressed proteins with antiapoptotic (BHRF1, BALF1) and immunomodulatory (BCRF1, BARF1, BILF1, BGLF5, BNLF2a, BLLF3) functions promote viral production, keeping the cell intact because they inhibit apoptotic signals and promote immune evasion. In tumorigenesis, they could also play a role, since these genes are expressed in the abortive lytic cycle and continue to provide antiapoptotic capacity and promote cell transformation [164]. In addition, a very important point is the latency establishment in EBV-mediated epithelial tumors. Considering that in both NPC and EBVaGCs, a clonal latent EBV is systematically detected in tumors and that latency is hardly found in non-tumor epithelial, an important concern is to know what factors promote EBV latency establishment in epithelial cells. In this respect, it has been suggested that previous DNA damage can favor the possibility of EBV latency establishment [53]. Thus, in periodontitis and during *P. gingivalis* infection, the inflammatory microenvironment, oxidative stress, and subsequent DNA damage can favor the possibility of EBV latency in these altered cells [95,172].

In consequence, the evidence supports the participation of EBV and *P. gingivalis* in the development of oral cancer through the activation of oncogenic pathways, in addition to *P. gingivalis*-mediated EBV reactivation, which allows us to propose a model of interaction between both microorganisms, although more studies are warranted to confirm this hypothesis (Figure 2).

## 6. Conclusions and Remarks

EBV is etiologically associated with epithelial tumors such as undifferentiated NPCs and with a subset of gastric cancers. The potential role of EBV in oral cancers is unknown. Periodontitis is a dysbiotic disease characterized by a chronic inflammation caused by an imbalance between subgingival microbiota, mostly caused by the key pathogen *P. gingivalis* and the host immune response. *P. gingivalis* is related to periodontal disease when its relative abundance is increased. Moreover, it has been related to OSCCs due to its ability to generate an inflammatory environment and the activity of its virulence factors (fimbriae, LPS, gingipains, capsule) in the activation of oncogenic pathways. 

We suggest a bidirectional interaction between EBV-positive oral epithelial cells and *P. gingivalis*, given the DNA damage that the bacteria produce through activation of ROS, which would allow EBV to establish latency in epithelial cells, a requirement for the development of cancer. In addition, butyric acid, produced as a waste compound of the bacteria, would favor inhibition of deacetylase (HDACs) and EBV-BZL1 gene activation. The activation of the IE genes generates an EBV abortive lytic cycle with the expression of some lytic genes, including BARF1. BARF1 is considered an oncogene present in carcinomas and not in lymphomas, with potential importance in EBV-associated oral cancers. On the other hand, EBV infection causes immunosuppression, which favors the immunological evasion of *P. gingivalis* and facilitates the increase in its relative abundance. Finally, it has been described that both microorganisms on their own have the ability to activate cell signaling pathways that promote tumorigenesis, suggesting the necessity to deepen the study of the interaction between both in the development of oral cancer. In summary, we propose an interaction mechanism in which *P. gingivalis*, by promoting inflammation, alters either the establishment of EBV latency or the expression of abortive lytic genes involved in the oncogenic potential of EBV (Figure 2); however, experimental studies are required to verify this hypothesis.

## Figures and Tables

**Figure 1 pathogens-09-01059-f001:**
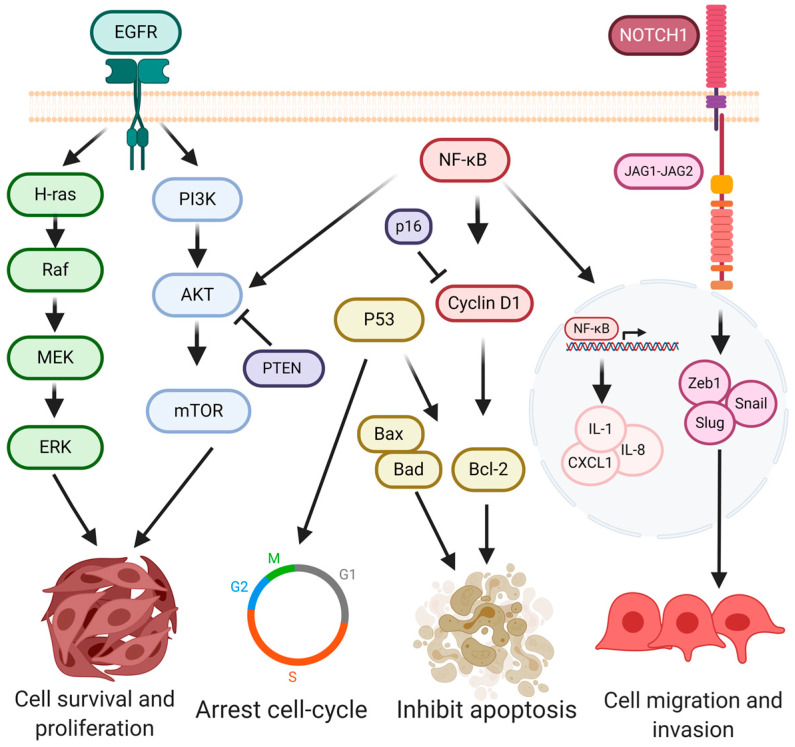
Altered pathways in oral tumorigenesis. Mutations or alteration in proteins or transcription factors involved in the deregulation of substantial pathways for cell survival and maintenance: proliferation, cycle-cell, apoptosis causing the presence of tumor characteristics in the cell-like migration.

**Figure 2 pathogens-09-01059-f002:**
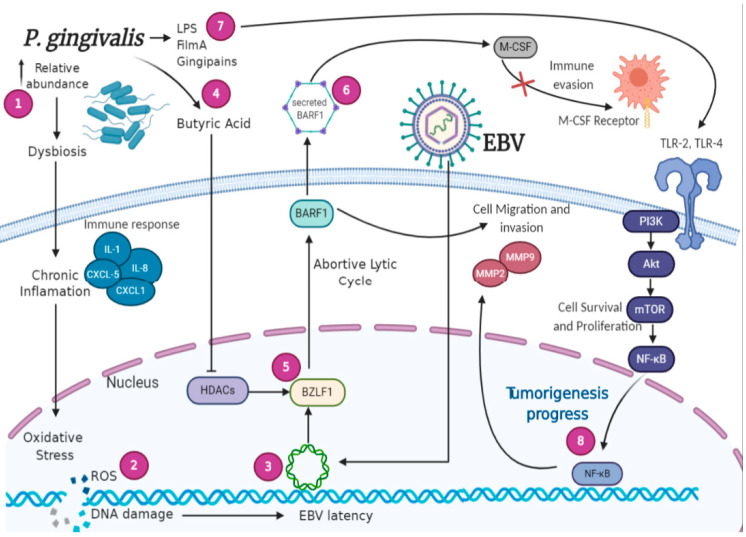
A suggested interaction model between EBV and *P. gingivalis* in the development of oral cancer. (1) The increase in the relative abundance of *P. gingivalis* leads to a dysbiotic oral microenvironment, in turn promoting chronic inflammation. (2) Products of oxidative stress are reactive oxygen species (ROS), which can lead to DNA damage. (3) DNA alterations favor the establishment of EBV latency in oral epithelial cells. (4, 5) Butyric acid wasted by *P. gingivalis* inhibits HDACs, promotes histone acetylation, EBV IE gene expression, which in turn promotes the abortive lytic cycle activation in which some latent and lytic viral genes are expressed, such as BARF1. (6) The secreted protein BARF1 modulates host immune response, through M-CSF acting as a decoy receptor, promoting a reduction in macrophage activation and differentiation. (7) *P. gingivalis*, through its virulence factors, can activate EGFR/PI3K/Akt/mTOR pathway (8) The sum of these factors and the inflammatory environment caused by host immune response contribute to the transformation process and development of oral cancer.

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
