# Peer review of "Epstein–Barr Virus—Oral Bacterial Link in the Development of Oral Squamous Cell Carcinoma"

_pathogens, 2020, doi:10.3390/pathogens9121059_

Round 1
Reviewer 1 Report
In the manuscript entitled: “Epstein Barr virus—oral bacterial link in the development of oral squamous cell carcinoma”, the authors identified the mechanisms by which EBV and P gingivalis independently or synergistically can collaborate for oral carcinogenesis.
The authors found that EBV is etiologically associated with epithelial tumors such as undifferentiated NPCs and with a subset of gastric cancers. A potential role of EBV in oral cancers is unknown. Periodontitis is a dysbiotic disease characterized by a chronic inflammation caused by an imbalance between subgingival microbiota, mostly caused by the key pathogen P. gingivalis and the host immune response. P. gingivalis is related to periodontal disease when its relative abundance is increased.
The authors concluded that both microorganisms on their own have the ability to activate cell signaling pathways that promote tumorigenesis, suggesting the necessity to deepen the study of the interaction between both in the development of oral cancer.
Major comments:
In general, the idea and innovation of this study, regards the virus and gingival bacteria expression during oral cancer is interesting, because the role of biomarkers released during oral diseases is validated but further studies on this topic could be an innovative issue in this field could be open a creative matter of debate in literature by adding new information. Moreover, there are few reports in the literature that studied this interesting topic with this kind of study design.
The study was well conducted by the authors; However, there are some concerns to revise that are described below.
The introduction section resumes the existing knowledge regarding the important factor linked with oral cancer and periodontal bacteria.
However, as the importance of the topic, the reviewer strongly recommends, before a further re-evaluation of the manuscript, to update the literature through read, discuss and must cites in the references with great attention all of those recent interesting articles, that helps the authors to better introduce and discuss the role of some others mediators, such as ADMA and suPAR and A.Actino involved during oral diseases and periodontitis: 1) Isola G, Alibrandi A, Currò M, Matarese M, Ricca S, Matarese G, Ientile R, Kocher T. Evaluation of salivary and serum ADMA levels in patients with periodontal and cardiovascular disease as subclinical marker of cardiovascular risk. J Periodontol. 2020 91;8: 1076-1084. doi: 10.1002/JPER.19-0446. 2) Isola G, Polizzi A, Alibrandi A, Williams RC, Leonardi R. Independent impact of periodontitis and cardiovascular disease on elevated soluble urokinase-type plasminogen activator receptor (suPAR) levels. J Periodontol. 2020 Oct 22. doi: 10.1002/JPER.20-0242. 3) Isola G, Polizzi A, Patini R, Ferlito S, Alibrandi A, Palazzo G. Association among serum and salivary A. actinomycetemcomitans specific immunoglobulin antibodies and periodontitis. BMC Oral Health. 2020 Oct 15;20(1):283. doi: 10.1186/s12903-020-01258-5.
The authors should be better specified, at the end of the introduction section, the rational of the study and the aim of the review. In the material and methods section, should better clarify the role of A.Actino and the role of ADMA and suPAR during periodontitis that could help to accelerate OSCC appearance.
The discussion section appears well organized with the relevant paper that support the conclusions, even if the authors should better discuss the relationship between periodontitis and oral cancer. The conclusion should reinforce in light of the discussions.
In conclusion, I am sure that the authors are fine clinicians who achieve very nice results with their adopted protocol. However, this study, in my view requests a revision before a futher re-evaluation of the manuscript.
Minor Comments:
Abstract:
- Better formulate the abstract section by better describing the aim of the study
Introduction:
- Please refer to major comments
Discussion
- Please add a specific sentence that clarifies the results obtained in the first part of the discussion
- Page 7 last paragraph: Please reorganize this paragraph that is not clear
Author Response
1.- Better formulate the abstract section by better describing the aim of the study.
Response: Considering your comment, the abstract has been modified to better describe the aim of the review in lines 11 to 26.
2.- The authors should be better specified, at the end of the introduction section, the rational of the study and the aim of the review.
Response: We appreciate your comment. For that, a sentence describing the aim of the review was added in line 61.
3.- Update the literature through read, discuss and must cites in the references with great attention all of those recent interesting articles, that helps the authors to better introduce and discuss the role of some others mediators, such as ADMA and suPAR and A.Actino involved during oral diseases and periodontitis: 1) Isola G, Alibrandi A, Currò M, Matarese M, Ricca S, Matarese G, Ientile R, Kocher T. Evaluation of salivary and serum ADMA levels in patients with periodontal and cardiovascular disease as subclinical marker of cardiovascular risk. J Periodontol.2020 91;8: 1076-1084. doi: 10.1002/JPER.19-0446. 2) Isola G, Polizzi A, Alibrandi A, Williams RC,Leonardi R. Independent impact of periodontitis and cardiovascular disease on elevated soluble urokinase-type plasminogen activator receptor (suPAR) levels. J Periodontol. 2020 Oct 22. doi:10.1002/JPER.20-0242. 3) Isola G, Polizzi A, Patini R, Ferlito S, Alibrandi A, Palazzo G.Association among serum and salivary A. actinomycetemcomitans specific immunoglobulin antibodies and periodontitis. BMC Oral Health. 2020 Oct 15;20(1):283. doi: 10.1186/s12903-020-01258-5.
Response: Thank you for the suggested articles, they were included and discussed in page 4 line 167 to 185.
4.- Page 7 last paragraph: Please reorganize this paragraph that is not clear
Response: I appreciate your comment, the paragraph has been rewritten more clearly in the line to 329 to 339.
5.- The discussion section appears well organized with the relevant paper that support the conclusions, even if the authors should better discuss the relationship between periodontitis and oral cancer. The conclusion should reinforce in light of the discussions. Please add a specific sentence that clarifies the results obtained in the first part of the discussion.
Response: In line 187 to 200, the relationship between periodontitis and oral cancer was better discussed and a sentence was added clarifying the result of the review in line 363 and in line 401.
Reviewer 2 Report
In this review, authors showed the collaboration of Epstein Barr virus (EBV) and oral bacterium Porphyromonas gingivalis in the development and progression of oral squamous cell carcinoma. This is logical and interesting. The reviewer feels the authors should correct a typo (TEM to EMT, line 168).
Author Response
1.- The reviewer feels the authors should correct a typo (TEM to EMT, line 168).
Response: The correction was made in line 211
Round 2
Reviewer 1 Report
The authors have well addressed to all reviewer's comments. I suggest the acceptance of this interesting manuscript.